

# Comparative genomics of 16 *Microbacterium* spp. that tolerate multiple heavy metals and antibiotics

Deric R. Learman, Zahra Ahmad, Allison Brookshier,
Michael W. Henson, Victoria Hewitt, Amanda Lis, Cody Morrison,
Autumn Robinson, Emily Todaro, Ethan Wologo, Sydney Wynne,
Elizabeth W. Alm and Peter S. Kourtev

Institute for Great Lakes Research and Department of Biology, Central Michigan University,
Mount Pleasant, MI, USA

## ABSTRACT

A total of 16 different strains of *Microbacterium* spp. were isolated from contaminated soil and enriched on the carcinogen, hexavalent chromium [Cr(VI)]. The majority of the isolates (11 of the 16) were able to tolerate concentrations (0.1 mM) of cobalt, cadmium, and nickel, in addition to Cr(VI) (0.5–20 mM). Interestingly, these bacteria were also able to tolerate three different antibiotics (ranges: ampicillin 0–16 μg ml$^{-1}$, chloramphenicol 0–24 μg ml$^{-1}$, and vancomycin 0–24 μg ml$^{-1}$). To gain genetic insight into these tolerance pathways, the genomes of these isolates were assembled and annotated. The genomes of these isolates not only have some shared genes (core genome) but also have a large amount of variability. The genomes also contained an annotated Cr(VI) reductase (*chrR*) that could be related to Cr(VI) reduction. Further, various heavy metal tolerance (e.g., Co/Zn/Cd efflux system) and antibiotic resistance genes were identified, which provide insight into the isolates' ability to tolerate metals and antibiotics. Overall, these isolates showed a wide range of tolerances to heavy metals and antibiotics and genetic diversity, which was likely required of this population to thrive in a contaminated environment.

Corresponding author
Deric R. Learman,
deric.learman@cmich.edu

# INTRODUCTION

Heavy metals, while naturally occurring, cause harm to both human and ecosystem health. Specifically, chromium is a mutagen and carcinogen and its presence in the environment can be natural due to anthropogenic activities, such as industrial manufacturing (e.g., metal plating and tanneries) and mining (chromite ore) (*Barak et al., 2006*; *Brose & James, 2010*; *Cheng, Holman & Lin, 2012*). In the environment, chromium can exist as hexavalent chromium [Cr(VI)], which is soluble and more toxic, or as insoluble trivalent chromium [Cr(III)] (*Bartlett, 1991*; *Cheng, Holman & Lin, 2012*). Thus, the redox cycling of chromium is critical to understanding its impacts on the environment. In natural systems, Cr(III) can be oxidized by manganese oxides and hydrogen peroxide,

while Cr(VI) can be reduced by ferrous iron and hydrogen sulfides (*Brose & James, 2010*; *Oze et al., 2004*; *Viti et al., 2013*).

Many bacteria are able to tolerate Cr(VI) stress based on their ability to transport it outside the cell or enzymatically reduce it to the less toxic form Cr(III). The efflux pump, *chrA*, has been associated with providing Cr(VI) resistance by transporting it outside the cell (*Cervantes & Ohtake, 1988*; *Cervantes et al., 1990*; *Nies, Nies & Silver, 1990*). A 2008 study found 135 *chrA* orthologs of this efflux pump (*Ramirez-Diaz et al., 2008*); however, the presence of *chrA* is not always a sole predictor of the amount of Cr(VI) that bacteria can resist (*Henne et al., 2009a*). Bacteria can also have additional efflux pumps that provide resistance to other metals (*Nies, 2003*; *Silver, 1996*), which is advantageous as anthropogenically impacted sites are often contaminated with multiple stressors. Further, there is a well-known association between metal and antibiotic tolerance (*Baker-Austin et al., 2006*; *Seiler & Berendonk, 2012*; *Wright et al., 2006*). For example, *Staphylococcus* species use multiple efflux pumps to tolerate chromium, lead, and penicillin (*Ghosh et al., 2000*) and *Salmonella abortus equi* strains were able to tolerate chromium, cadmium, mercury, and ampicillin (*Ug & Ceylan, 2003*). Thus, efflux pumps can play a vital role in how bacteria thrive in contaminated ecosystems.

Bacteria are also known to catalyze the reduction of Cr(VI) aerobically. Specifically, certain bacteria use a soluble NADH/NADPH-dependent oxidoreductase to reduce Cr(VI) (*Ackerley et al., 2004a*; *Barak et al., 2006*; *Cheung & Gu, 2007*; *Gonzalez et al., 2005*; *Park et al., 2000*). Two well-studied Cr(VI) reductases are *chrR* and *yieF*. In *Pseudomonas putida*, *chrR* reduces Cr(VI) via the transfer of one electron, generating Cr(V), a reactive intermediate, which then requires a second electron transfer to generate the more stable insoluble Cr(III) (*Park et al., 2000*). Conversely, *Escherichia coli* uses *yieF* to catalyze the transfer of four electrons to reduce Cr(VI) (*Ackerley et al., 2004b*; *Barak et al., 2006*; *Ramirez-Diaz et al., 2008*).

The objective of this study was to examine the genomes of 16 *Microbacterium* spp. strains isolated from metal contaminated soil to identify their genetic potential and physiological ability to reduce Cr(VI) and to tolerate both heavy metals and antibiotics. A recent genomic study of four Cr(VI) reducing *Microbacterium* spp. identified two putative reductases (*Henson et al., 2015*), one genetically similar to *chrR* in *Thermus scotoductus* (*Opperman, Piater & van Heerden, 2008*) and the other to *yieF* of *Arthrobacter* sp. RUE61a (*Niewerth et al., 2012*); however, these putative reductases had low sequence similarity to the well-studied reductases found in *Pseudomonas putida* and *E. coli*. While other studies have shown *Microbacterium* isolates are able to reduce Cr(VI) (*Humphries et al., 2005*; *Liu et al., 2012*; *Pattanapipitpaisal, Brown & Macaskie, 2001*; *Soni et al., 2014*), there are still unknowns related to the genetics of Cr(VI) reduction and resistance. Further, the study by *Henson et al. (2015)* documented high levels of genomic variability between the isolates, which might be related to ecotypes. However, the conclusions of this study were limited based on the low number of genomes examined. Comparative genomics has been used in various environments and microorganisms to examine inter-strain variation and ecotypes (*Briand et al., 2009*; *Coleman & Chisholm, 2010*; *Denef et al., 2010*; *Frangeul et al., 2008*; *Humbert et al., 2013*;

*Martiny, Coleman & Chisholm, 2006*; *Meyer & Huber, 2014*; *Meyer et al., 2017*). Thus, this study examined the genomic and physiological variability of heavy metal and antibiotic resistance of 16 *Microbacterium* strains from the same contaminated soil environment to elucidate the potential genomic flexibility of closely related strains.

## METHODS

### Bacterial isolation

Isolation of *Microbacterium* spp. from soil is described in *Kourtev, Nakatsu & Konopka (2009)* and Henson et al. (*Henson et al., 2015*). Bill Jervis from the Indiana Department of Transport provided site access (no permit was required) and the project did not involve endangered or protected species. Briefly, contaminated soil samples were collected from the Department of Transportation site in Seymour, IN, USA. Previous studies have documented the site was contaminated with Pb (1,156 $\mu g^{-1}$ soil), Cr (5,868 $\mu g^{-1}$ soil), and hydrocarbons (toluene and xylenes: >200 $\mu g\ g^{-1}$ soil) (*Joynt et al., 2006*; *Kourtev, Nakatsu & Konopka, 2006*; *Nakatsu et al., 2005*). Bacteria were initially isolated from the soil on 50% tryptic soy agar amended with 0.25 mM Cr(VI) ($K_2CrO_4$). From this, 16 isolates were further selected based on their ability to grow on at least 0.5 mM Cr(VI). The isolates were stored in glycerol stocks at $-80\ °C$ until further use.

### Cr(VI) reduction

For Cr(VI) reduction experiments, isolates were grown in 25% tryptic soy broth (TSB) with 0.5 or 1 mM Cr(VI) at 30°C and 225 rpm for 72 h. Two concentrations were used as certain isolates had relatively lower growth at 1 mM compared to other isolates. The cultures were pelleted and Cr(VI) reduction was determined using a colorimetric assay (*Henson et al., 2015*; *Urone, 1955*).

### DNA sequencing and bioinformatics

Isolates were grown in TSB with 0.5 or 2 mM Cr(VI) at 30 °C and 225 rpm for 24–72 h. The cultures were pelleted via centrifugation and stored at $-20\ °C$ until used in DNA extractions. FastDNA Spin Kits (MP Biomedical) were used to extract genomic DNA from the thawed pellets. The resulting DNA was stored at $-20\ °C$. The DNA from the 16 bacterial isolates were sent to Cincinnati Children's Hospital Medical Center's Genetic Variation and Gene Discovery Core facility for whole genome shotgun sequencing using one lane of an Illumina HiSeq 2000 (100bp PE). The resulting raw genomic reads can be found in the National Center for Biotechnology Information's (NCBI) Short Read Archives (SRA), accession number: SRP120551.

Raw reads were evaluated for quality with FastQC (https://www.bioinformatics.babraham.ac.uk/projects/fastqc/) and trimmed and quality filtered with Trimmomatic (v 0.33) (*Bolger, Lohse & Usadel, 2014*). Sequence quality scores were low on R2 (assessed by FastQC), so the trimmed reads were cut (length of 70bp) and passed through another quality filter (-Q33 -q 30 -p 50) with FastX toolkit (http://hannonlab.cshl.edu/fastx_toolkit/). Reads for each sample were subsampled to four million reads (multiple subsampling depths were tested) with seqtk (https://github.com/lh3/seqtk).

Using multiple different read depths, MEGAhit (*Li et al., 2016*) was used to assemble the data, which were then assessed for quality with QUAST (*Gurevich et al., 2013*). CheckM (*Parks et al., 2015*) was used to examine genome completeness and contamination (taxon marker *Microbacterium*). The resultant genomes were annotated by the Department of Energy's Joint Genome Institute Integrated Microbial Genomes (IMG) system (*Markowitz et al., 2012*) and are publicly available (see Table S1). A pangenomic analysis of 20 genomes (from this study and *Henson et al., 2015*) was conducted using the An'vio program (version 3) (*Eren et al., 2015*) following the pangenomics workflow from *Delmont & Eren (2018)*. The pangenomic analysis within An'vio also utilized other programs like HMMER (*Eddy, 2011*), Prodigal (*Hyatt et al., 2010*), and NCBI's blastp (*Altschul et al., 1997*). An'vio was also used to define clusters of orthologous groups (COGs) for the pangenomic analysis using the command *anvi-run-ncbi-cogs*.

All 16, 16S rRNA gene sequences from the Cr(VI) isolates were downloaded from IMG. These genes were used to generate a maximum likelihood phylogenetic tree in MEGA7 (*Kumar, Stecher & Tamura, 2016*) and evolutionary history was inferred using the maximum likelihood Tamura–Nei model (*Tamura & Nei, 1993*). Within IMG, annotations were searched for genes related to Cr(VI) resistance and Cr(VI) reduction via keywords (e.g., efflux, reductase, and chromate). Annotated Cr(VI) reductase genes were compared via BLASTP (one-way comparison on NCBI's website) to a known Cr(VI) reducing *chrR* from *Pseudomonas putida* KT2440 (NP_746257) (*Ackerley et al., 2004b*). Genes related to heavy metal resistance and antibiotic resistance were searched using functional categories. Beta-lactam, chloramphenicol, and vancomycin resistance genes were also identified using KEGG pathway KO identifiers. The COG category "Inorganic ion transport and metabolism" was used to find annotated heavy metal efflux genes and metal resistance genes.

## Metal and antibiotic tolerance assays

Bacterial isolates were grown on agar plates (10% TSB, 1.5% agar, and 0.5 mM Cr(VI)) at 30 °C until colonies were visible. Colonies were then streaked onto fresh agar plates amended with different concentrations of heavy metals (Cd 0–5 mM, Co 0–5 mM, Cr 0–20 mM, Cu 0–1 mM, Ni 0–15 mM, and Zn 0–1 mM). Duplicate plates were incubated at 30 °C for 14 days and growth was confirmed by the presence of visible colonies. All bacterial isolates were plated on each metal concentration in duplicate.

The Minimum Inhibitory Concentration (MIC) for ampicillin, chloramphenicol, and vancomycin was tested on each isolate using MIC test strips (Liofilchem®). To do this, each of the 16 *Microbacterium* spp. isolates was streaked onto agar plates (50% TSB and 1.5% agar) with 0.5 mM Cr and incubated at 30 °C for 5 days. Next, each isolate was suspended in a 0.85% sterile saline solution until it reached a density that approximated the 0.5 McFarland Turbidity Standard. The isolates were then spread uniformly on a Mueller–Hinton agar plate. A MIC test strip was placed in the center of each plate. The plates were incubated at 30 °C for 24 h. After the 24 h, the plates were removed and the MIC of each isolate to each antibiotic was determined by visually documenting the zones of inhibition.

**Table 1 Heavy metal tolerance and chromate reduction.**

| Isolate | Cr | Ni | Co | Cd | Zn | Cu | Cr reduction (%)* |
|---------|-----|-----|-----|-----|-----|-----|-----|
| *Microbacterium* sp. A20 | 20.0 | 1.0 | 0.1 | 0.1 | 1.0 | 1.0 | 41.6 |
| *Microbacterium* sp. K19 | 2.0 | 1.0 | 0.1 | 0.1 | 1.0 | 1.0 | 34.0 |
| *Microbacterium* sp. K21 | 2.0 | 1.0 | 0.1 | 0.1 | 1.0 | 1.0 | 39.3 |
| *Microbacterium* sp. K22 | 2.0 | 1.0 | 0.1 | 0.1 | 0.1 | 1.0 | 42.9 |
| *Microbacterium* sp. K24 | 20.0 | 1.0 | 1.0 | 0.0 | 1.0 | 1.0 | 41.6 |
| *Microbacterium* sp. K27 | 2.0 | 1.0 | 1.0 | 0.1 | 0.1 | 1.0 | 83.0 |
| *Microbacterium* sp. K2B2 | 0.5 | 0.1 | 0.0 | 0.1 | 0.1 | 0.1 | 36.0 |
| *Microbacterium* sp. K30 | 20.0 | 1.0 | 1.0 | 0.1 | 1.0 | 1.0 | 41.6 |
| *Microbacterium* sp. K31 | 2.0 | 0.1 | 0.1 | 0.1 | 1.0 | 1.0 | 55.3 |
| *Microbacterium* sp. K33 | 20.0 | 1.0 | 0.1 | 0.1 | 1.0 | 1.0 | 88.8 |
| *Microbacterium* sp. K35 | 2.0 | 0.1 | 0.1 | 0.1 | 0.1 | 1.0 | 3.6 |
| *Microbacterium* sp. K36 | 0.5 | 1.0 | 0.1 | 0.1 | 0.1 | 1.0 | 10.7 |
| *Microbacterium* sp. K40 | 0.5 | 1.0 | 0.1 | 0.1 | 1.0 | 1.0 | 30.3 |
| *Microbacterium* sp. K41 | 2.0 | 1.0 | 0.1 | 0.0 | 0.1 | 1.0 | 28.1 |
| *Microbacterium* sp. K5D | 2.0 | 1.0 | 0.1 | 0.0 | 0.1 | 1.0 | 0.0 |
| *Microbacterium* sp. PF3 | 20.0 | 1.0 | 0.1 | 0.0 | 1.0 | 1.0 | 50.6 |

**Metal tolerance (mM)**

Notes:
Heavy metal (cadmium, chromium, cobalt, copper, nickel, and zinc) tolerance (in mM) of each isolate and chromate reduction (%) data.
* Reduction % after 72 h with one mM Cr except for K19, 31, 27 which was 0.5 mM.
Data shown in percentages since two chromate concentrations were used.

# RESULTS AND DISCUSSION

## Isolation of Cr(VI) reducing bacteria

Though originally isolated and studied for their ability to reduce Cr(VI) (*Henson et al., 2015*; *Kourtev, Nakatsu & Konopka, 2009*), it was speculated that these isolates may also be tolerant to other contaminants, because the soils were contaminated with other heavy metals and organic solvents (*Joynt et al., 2006*; *Kourtev, Nakatsu & Konopka, 2006*; *Nakatsu et al., 2005*). Among the 16 isolates studied here, a wide range of Cr(VI) tolerance and reducing ability was found (Table 1). The majority (13) of the isolates could tolerate two mM Cr(VI), while three of the isolates (K2B2, K36, and K40) could only tolerate 0.5 mM Cr(VI). Interestingly, five of the 13 isolates that were able to tolerate two mM Cr(VI) (A20, K24, K30, K33, and PF3) had observable colonies on plates containing up to 20 mM Cr(VI). Further, the 16 isolates also showed a wide range in Cr(VI) reduction (0–88.8%) over a 72-h incubation (Table 1). Only two isolates, K27 and K33, reduced over 80% of the Cr(VI) in the experiment. Interestingly, K27 reduced 83% of the Cr(VI) in liquid medium but was only able to tolerate 0.5 mM Cr(VI) when grown on solid medium, while K33 tolerated up to 20 mM Cr(IV) and reduced 88% of the Cr(IV) present. Two additional isolates, K31 and PF3, reduced over 50% of the Cr(VI) in the experiment. There was no statistical relationship between the ability of isolates to tolerate and their ability to reduce Cr(VI).

**Table 2** Assembly statistics and quality assurance data.

| Isolate | Total length (Mb) | GC (%) | Number of contigs | Largest contig (bp) | N50 (bp) | Est. Sequencing coverage | Comp. (%) | Cont. (%) | Strain heterogeneity (%) |
|---|---|---|---|---|---|---|---|---|---|
| *Microbacterium* sp. A20 | 3.94 | 68.53 | 75 | 436,954 | 166,569 | 142 | 99.06 | 0.54 | 0 |
| *Microbacterium* sp. K19 | 3.89 | 68.69 | 54 | 479,964 | 187,157 | 144 | 99.44 | 1.01 | 20 |
| *Microbacterium* sp. K21 | 3.85 | 68.33 | 34 | 537,739 | 330,631 | 146 | 99.06 | 0.18 | 0 |
| *Microbacterium* sp. K22 | 3.94 | 68.52 | 96 | 258,133 | 83,317 | 142 | 99.06 | 0.71 | 33.33 |
| *Microbacterium* sp. K24 | 4.23 | 68.90 | 142 | 248,507 | 56,118 | 132 | 99.35 | 3.93 | 9.09 |
| *Microbacterium* sp. K27 | 3.75 | 68.51 | 37 | 342,064 | 187,856 | 149 | 98.66 | 0.18 | 0 |
| *Microbacterium* sp. K2B2 | 3.94 | 68.52 | 76 | 436,909 | 120,725 | 142 | 99.06 | 0.54 | 0 |
| *Microbacterium* sp. K30 | 4.06 | 69.16 | 96 | 295,140 | 88,745 | 138 | 98.39 | 2.61 | 0 |
| *Microbacterium* sp. K31 | 3.77 | 68.43 | 39 | 443,452 | 183,886 | 148 | 98.66 | 0.18 | 0 |
| *Microbacterium* sp. K33 | 3.89 | 68.68 | 110 | 311,089 | 66,684 | 144 | 99.44 | 1.49 | 0 |
| *Microbacterium* sp. K35 | 3.62 | 70.91 | 379 | 168,068 | 17,080 | 155 | 96.70 | 1.82 | 12.5 |
| *Microbacterium* sp. K36 | 3.31 | 70.53 | 91 | 183,861 | 73,738 | 169 | 98.10 | 0.60 | 0 |
| *Microbacterium* sp. K40 | 3.77 | 68.40 | 45 | 442,332 | 195,355 | 148 | 99.06 | 0.09 | 0 |
| *Microbacterium* sp. K41 | 3.62 | 70.75 | 454 | 89,969 | 15,114 | 155 | 97.62 | 1.37 | 0 |
| *Microbacterium* sp. K5D | 3.80 | 68.35 | 49 | 442,284 | 195,368 | 147 | 99.06 | 0.09 | 0 |
| *Microbacterium* sp. PF5 | 3.74 | 70.80 | 144 | 125,065 | 47,267 | 150 | 97.23 | 1.82 | 0 |

**Note:**
Genome assembly statistics (total length, GC%, and number of contigs) and quality assurance data (completeness, contamination, and strain heterogeneity).

## Genomic assembly and pangenomic analysis

Previous research has shown Cr(VI) tolerance to be related to a resistance gene (*chrA*) (*Cervantes & Ohtake, 1988*; *Cervantes et al., 1990*) and/or Cr(VI) reductases (*chrR* and *yieF*; (*Ackerley et al., 2004b*; *Barak et al., 2006*; *Park et al., 2000*; *Ramirez-Diaz et al., 2008*). To gain a better understanding of the Cr(VI) tolerance of these isolates, their genomes were assembled and annotated. The isolate genomes ranged in total length from 3.31 to 4.23 Mb and all had GC content ranging from 68 to 70%% (Table 2). A genomic study of 10 *Macrobacterium* sp. documented similar GC ranges and genome length (*Corretto et al., 2015*). The assembled genomes were generally over 96.7% complete and all genomes had under 4% contamination (Table 2).

Phylogenetic analysis of the 16S rRNA gene indicated that all of the isolates belong to the genus *Microbacterium* (Fig. S1; Table S2). Specifically, the isolates were closely related to *Microbacterium oxydans*, *Microbacterium maritypicum*, or *Microbacterium paraoxydans*. *Microbacterium* spp. have been isolated from sites contaminated with metals (*Avramov et al., 2016*; *Bollmann et al., 2010*; *Corretto et al., 2015*), radionuclides (*Nedelkova et al., 2007*), and petroleum (*Avramov et al., 2016*; *Chauhan et al., 2013*; *Wang et al., 2014*). Moreover, some environmental isolates, such as *Microbacterium laevaniformans* strain OR221, have shown to tolerate multiple metals (Ni, Co, and Cd) (*Bollmann et al., 2010*). A genomic analysis of the latter strain provided evidence of genes (e.g., transporter and detoxification genes) that could aid the strain's ability to tolerate metals (*Brown et al., 2012*). Other *Microbacterium* spp. have been known to
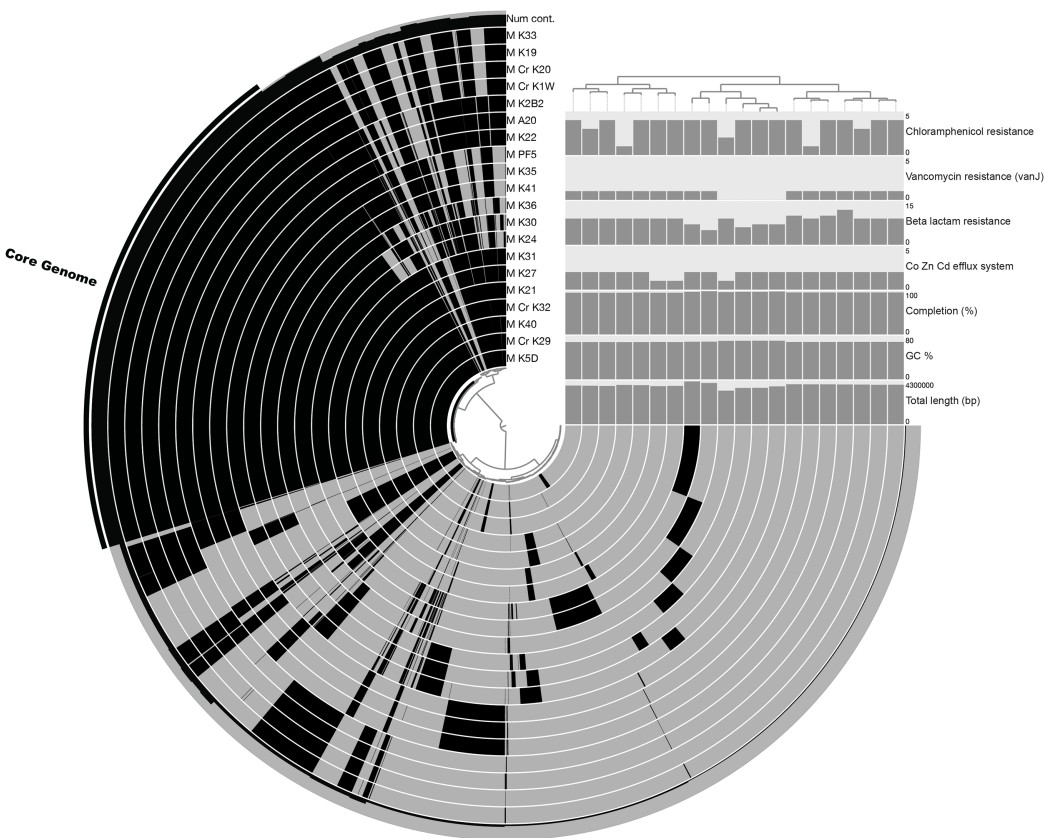

**Figure 1** **Pangenomic analysis of heavy metal tolerant *Microbacterium spp.*** The figure was made in An'vio with the items order in presence/absence (D: Euclidean; L: Ward) and the samples ordered by PC frequency. The outer most complete ring (labeled Num cont.) represents the number of genomes that a gene occurs in, with the core genome highlighted with an additional black bar. The internal rings represent an isolate's genome with the black color representing the presence of genes in a genome (gray is the absence of a gene). The bar charts show quality control measurements (total genome length [base pairs], GC content, and percent completion) and also the abundance of genes that were annotated as Co, Zn, Cd efflux system genes and antibiotic tolerance genes (beta lactam, vancomycin, and chloramphenicol resistance).

reduce Cr(VI): *Microbacterium* sp. SUCR140 (*Soni et al., 2014*), *Microbacterium* sp. chr-3 (*Focardi, Pepi & Focardi, 2013*), and *Microbacterium* sp. CR-07 (*Liu et al., 2012*).

The pan-genome size of the 20 isolates was 7,902 protein clusters with the core genome encompassing 2,073 protein clusters (26%) (Fig. 1). The majority of the genes in the core genome of the 20 isolates (36%) were placed in the "Metabolism" COG (Fig. 2). Genes in the COG categories "Cellular processes and signaling" and "Information storage and processing" comprised 21% and 17%, respectively of the core genome (Fig. 2). The flexible pangenome was dominated by genes in the COG categories "Metabolism" and "Poorly characterized/No assigned COG" (36% for both) (Fig. 2). Relative to the core pangenome, the flexible pangenome had few genes in the COG categories "Cellular processes and signaling (21%)" and "Information storage and processing 17%" (Fig. 2). The percentage of core genes in a genome can vary from 3% to 84% (*McInerney, McNally & O'Connell, 2017*). For example, the *Bacillus cereus* core genome is 27% of the

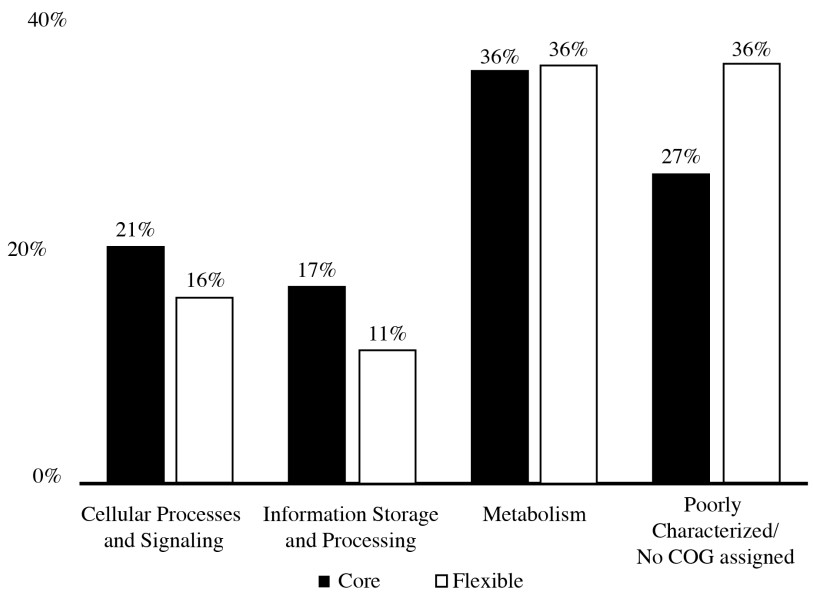

**Figure 2 COGs of share genes.** COG distribution of genes found in the core and flexible genomes of 20 *Microbacterium* isolates isolated from heavy metal contaminated soil.

pangenome, whereas the core genome of *Bacillus anthracis* is 65% (*McInerney, McNally & O'Connell, 2017*). An analysis of 20 *Microcystis* sp. showed the genome was comprised of 34–49% core genes and 51–66% flexible genes (*Meyer et al., 2017*). A study on the genomes of 12 *Prochlorococcus* isolates found a core genome ranging from 40% to 67% (*Kettler et al., 2007*). The relatively large genomic variability seen with the *Microbacterium* spp. examined here could be due to selective pressures that drove gene loss or horizontal gene transfer, which in the end enhanced the ability of the isolates to survive in a contaminated sediment environment.

The COG categories "Inorganic ion transport and metabolism (P)" and "Self-defense mechanism (V)" were examined due to their relation to heavy metal and antibiotic transport. The flexible genome has relatively higher percentages of genes in both the self-defense (1% of the core genome and 3% of the flexible genome) and inorganic ion transport and metabolism (5% of the core genome and 6% of the flexible genome) COG category. Each of these COG categories includes predicted gene functions for heavy metal transport and antimicrobial resistance and antibiotic efflux/transporters (Fig. 1). Interestingly, the core and flexible genomes contain different Co/Zn/Cd efflux system component genes and multidrug efflux pump genes, which suggest tolerance to heavy metals and antibiotics. The variability found here may be related to the ability of each isolate to tolerate different types and concentration of heavy metals and antibiotics.

## Genomic insights of Cr(VI) reduction

The annotated genomes from this study provided evidence that these isolates could reduce Cr(VI). All possess annotated Cr(VI) reductase, *chrR* (Table S3). In addition, *M.* sp K24 and K30 contained two annotated *chrR* genes. The Cr(VI) reductases were compared

with a known Cr(VI) reductase from *Pseudomonas putida* KT2440 (*Park et al., 2000*) and BLASTP results showed that all the *Microbacterium* spp. *chrR* genes shared a high degree of homology (ranging from 40% to 46% identity) (Table S3). Thus, it is likely that these *chrR* genes are responsible for the Cr(VI) reduction ability of these isolates. Interestingly, other *Microbacterium* spp. isolated from this site did not have annotated Cr(VI) reductases but BLAST searches were able to identify genes homologous to *chrR* and *yeiF* (*Henson et al., 2015*), further suggesting high interspecies genetic diversity in the putative *Microbacterium* spp. Cr(VI) reductases found in the same soil.

Cr(VI) tolerance might not be related to efflux pumps and Cr(VI) reductases. While the genomes did contain Cr(VI) specific reductases, they did not contain a Cr(VI) efflux pump (*chrA*). The lack of an assembled and annotated *chrA* was interesting since the ability of the isolates to reduce and resist Cr(VI) did not correlate. While the lack of an assembled *chrA* does not confirm its absence from the genome, tolerance could be related to other genes. Cr(IV) tolerance has been shown to be related genes that involve oxidative stress response (*Ackerley et al., 2004a*; *Cheng et al., 2009*), DNA repair (*Hu et al., 2005*; *Miranda et al., 2005*), and metabolism (*Brown et al., 2006*; *Decorosi et al., 2009*; *Henne et al., 2009b*).

## Tolerance to multiple heavy metals and antibiotics

The isolates' genomes all contained numerous genes that suggest these bacteria can tolerate multiple heavy metals (e.g., Co, Zn, and Cd) and antibiotics. When examining KEGG pathways that related to transport, all of the isolates had genes that were annotated to be Co/Zn/Cd efflux system components (Table S4). In addition, some isolates had a putative cadmium resistance protein (Table S4). KEGG pathways also documented genes that could provide tolerance to three different antibiotics (ampicillin, chloramphenicol, and vancomycin) (Table S4).

The isolates' physiological ability to tolerate heavy metals were examined to determine whether the inferred resistance based on the putative metal transport genes could be confirmed. All the isolates were streaked onto plates with various concentrations of Cd, Co, Cr, Cu, Ni, and Zn. Overall, 11 of the 16 isolates tolerated between 0.1 and 1.0 mM of all six tested metals (Table 1). All 16 isolates were able to tolerate Cr, Ni, Zn, and Cu (Table 1). As the isolates were enriched with Cr, it was expected that they would be able to tolerate higher concentrations of Cr. Overall, the bacteria isolated in this study were able to tolerate many heavy metals, a finding seen in other studies. *Microbacterium* spp. enriched from Ni-rich serpentine soils have been previously documented to tolerate multiple heavy metals including Cd (0.5–2.5 mM), Co (0.1–5 mM), Cr (1–5 mM), Cu (0.25–10 mM), Ni (5–15 mM), and Zn (5–10 mM) (*Abou-Shanab, Van Berkum & Angle, 2007*). Another study found that multiple isolates closely related to *M. oxydans* were able to tolerate Cu (0.25–16 mM), Cr (0.5–16 mM), and Ni (0.25–16 mM) (*Nedelkova et al., 2007*). As *Microbacterium* spp. have been isolated from contaminated environments, it is not surprising to find they can tolerate heavy metal contamination. While this study did not attempt to confirm the function of the Co/Zn/Cd efflux system components found in these genomes (Table S4), the presence of these genes could be related to the

**Table 3 Antibiotic minimum inhibitory concentrations.**

| Isolates | Amp* | Cam* | Van* |
|---|---|---|---|
| *Microbacterium* sp. A20 | 0.0 | 8.0 | 0.5 |
| *Microbacterium* sp. K19 | 3.0 | 12.0 | 3.0 |
| *Microbacterium* sp. K21 | 3.0 | 12.0 | 4.0 |
| *Microbacterium* sp. K22 | 1.5 | 1.5 | 1.0 |
| *Microbacterium* sp. K24 | 3.0 | 24.0 | 6.0 |
| *Microbacterium* sp. K27 | 0.0 | 12.0 | 4.0 |
| *Microbacterium* sp. K2B2 | 0.0 | 0.0 | 4.0 |
| *Microbacterium* sp. K30 | 3.0 | 16.0 | 3.0 |
| *Microbacterium* sp. K31 | 2.0 | 8.0 | 4.0 |
| *Microbacterium* sp. K33 | 2.0 | 16.0 | 3.0 |
| *Microbacterium* sp. K35 | 2.0 | 24.0 | 1.5 |
| *Microbacterium* sp. K36 | 3.0 | 24.0 | 1.5 |
| *Microbacterium* sp. K40 | 3.0 | 12.0 | 3.0 |
| *Microbacterium* sp. K41 | 1.0 | 1.5 | 1.5 |
| *Microbacterium* sp. K5D | 1.0 | 24.0 | 4.0 |
| *Microbacterium* sp. PF3 | 16.0 | 6.0 | 0.1 |

Notes:
Minimum inhibitory concentrations (μg/ml) of ampicillin, chloramphenicol, and vancomycin for each isolate.
* Ampicillin (Amp), chloramphenicol (Cam), and vancomycin (Van).

ability of these isolates to tolerate multiple heavy metals. Whether in culture (e.g., *Pseudomonas putida* or *Cupriavidus necator*, formerly *Alcaligenes eutrophus*) (*Manara et al., 2012*; *Nies, 1995*) or in soil microcosms (*Cabral et al., 2016*), cobalt-zinc-cadmium efflux system proteins have been shown to provide tolerance to heavy metals. Thus, it is possible that the presence of these genes plays the same role in the isolates documented in this study.

There is a long history of research linking metal tolerance and antibiotic resistance (*Baker-Austin et al., 2006*; *Calomiris, Armstrong & Seidler, 1984*; *Henriques et al., 2016*; *Seiler & Berendonk, 2012*). In this study, 13 of the 16 isolates showed tolerance to ampicillin, chloramphenicol, and vancomycin (Table 3). Specifically, 13 out of 16 isolates were able to grow in the presence of ampicillin with tolerances ranging from 1.5 to 16 μg ml$^{-1}$ (Table 3). Chloramphenicol tolerance was found in 15 out of 16 isolates, of which four isolates, K24, K35, K36, and K5D, tolerated 24 μg ml$^{-1}$ of the antibiotic (Table 3). *Microbacterium* isolates have been previously documented to tolerate various antibiotics. *Microbacterium* isolates from fish mucus have shown high antibiotic resistance to both ampicillin (>1,600 μg ml$^{-1}$) and chloramphenicol (> 960 μg ml$^{-1}$) (*Ozaktas, Taskin & Gozen, 2012*). Human isolated *Microbacterium* spp. have been reported to grow in the presence of vancomycin in concentrations ranging from 0.25 to 15 μg ml$^{-1}$ (*Gneiding, Frodl & Funke, 2008*). Other clinical *Microbacterium* isolates had MIC for ampicillin from 1 to 1.5 μg ml$^{-1}$ and vancomycin from 3 to 4 μg ml$^{-1}$ (*Laffineur et al., 2003*). A 2015 study found 26% of *Microbacterium* isolates were resistant to vancomycin (*Bernard & Pacheco, 2015*).

Metal and antibiotic co-tolerance has been reported in a number of environmentally isolated bacteria. Bacteria isolated from drinking water with noted antibiotic resistance were also tolerant to high levels of $Cu^{2+}$, $Pb^{2+}$, and $Zn^{2+}$ (*Calomiris, Armstrong & Seidler, 1984*). Environmental isolates from wastewater treatment plants have been shown to tolerate various heavy metals and antibiotics (*Shafique, Jawaid & Rehman, 2016*, *2017*). *Bacillus* spp. isolated from river water tolerated multiple heavy metals and antibiotics (*Shammi & Ahmed, 2016*). Co-tolerance to heavy metals, antibiotics, and polychlorinated biphenyls was also observed in isolates from Antarctic sediments (*Giudice et al., 2013*). Further, evidence of shared genetic tolerance mechanisms for metals and antibiotics was found in *Staphylococcus aureus* isolated from a polluted riverbank. *S. aureus* contained a novel *emrAB* operon encoding efflux pumps that were inducible by the heavy metals Cr(VI) and manganese and the antibiotics ampicillin and chloramphenicol (*Zhang et al., 2016*).

While this study does not prove function, the data suggest the antibiotic tolerances found in these isolates could be related to the antibiotic resistance genes documented in the genomes. For example, 15 out of 16 isolates were able to tolerate chloramphenicol matching the 15 genomes that contained an annotated chloramphenicol resistance gene (*cmlR*, *cmx*) (Table S4), a gene previously shown to provide *Corynebacterium striatum* with chloramphenicol tolerance (*Schwarz et al., 2004*; *Tauch et al., 1998*). It is possible *cmx* could therefore provide tolerance to the *Microbacterium* isolates. Similarly, all the *Microbacterium* isolates had an annotated class A beta-lactamase gene (*penP*), a gene that provides ampicillin tolerance to numerous bacteria (*Bush & Jacoby, 2010*), though not all isolates were resistant to ampicillin (Table 3).

## Implications to community functions

All *Microbacterium* spp. were isolated from the same contaminated soil and have highly similar 16S rRNA genes (99–100% BLAST identity). Yet, these isolates displayed both genomic variation and varying abilities to tolerate multiple heavy metals and antibiotics. Other studies have shown that closely related isolates can exhibit variable ranges of genomic and physiological differences (*Coleman & Chisholm, 2010*; *Henson et al., 2015*; *Hunt et al., 2008*; *Martiny, Coleman & Chisholm, 2006*; *Meyer & Huber, 2014*; *Qamar, Rehman & Hasnain, 2017*; *Rocap et al., 2003*; *Simmons et al., 2008*; *Welch et al., 2002*). A study of 78 *Myxococcus xanthus* isolates from a small soil plot found 21 different genotypes (*Vos & Velicer, 2006*). Thus, it is speculated that genomic and physiological diversity would allow populations to differentiate, potentially increasing resistance and/or resilience that would aid the community to thrive during metals and other contaminant stress. Further support for this could be found when examining the Cr(VI) reduction and resistance data. While none of the isolates had an annotated Cr(VI) resistance efflux pump (*chrA*), they all contained a Cr(VI) reductase (*chrR*), yet there was variability found in their ability to reduce and resist Cr(VI). As the isolates from this study were from a soil with multiple contaminants, specific isolates that more efficiently reduce Cr(VI) (making it less bioavailable) may allow other community members (without resistance genes) to thrive and possibly degrade other contaminants. Thus, the variation observed in

these isolates could be a reflection of how the environmental stressors (e.g., heavy metals) can promote genomic diversity and a more stable community.

## ACKNOWLEDGEMENTS

We thank Jorge W. Santo Domingo for assistance with data analysis and helpful comments on the manuscript. We also thank Amber Conley for assistance with lab work.
This is contribution number 114 of the CMU Institute for Great Lakes Research.

### Funding
Funding was provided by the CMU College of Science and Engineering and the Office of Research and Graduate Studies' Undergraduate Research and Creative Endeavors grant. The funders had no role in study design, data collection and analysis, decision to publish, or preparation of the manuscript.

### Grant Disclosures
The following grant information was disclosed by the authors:
CMU College of Science and Engineering.
Office of Research and Graduate Studies' Undergraduate Research Creative Endeavors grant.

### Competing Interests
The authors declare that they have no competing interests.

### Author Contributions

- Deric R. Learman conceived and designed the experiments, performed the experiments, analyzed the data, contributed reagents/materials/analysis tools, prepared figures and/or tables, authored or reviewed drafts of the paper, approved the final draft.
- Zahra Ahmad performed the experiments, approved the final draft.
- Allison Brookshier performed the experiments, approved the final draft.
- Michael W. Henson performed the experiments, analyzed the data, authored or reviewed drafts of the paper, approved the final draft.
- Victoria Hewitt performed the experiments, approved the final draft.
- Amanda Lis performed the experiments, approved the final draft.
- Cody Morrison performed the experiments, approved the final draft.
- Autumn Robinson performed the experiments, approved the final draft.
- Emily Todaro performed the experiments, approved the final draft.
- Ethan Wologo performed the experiments, approved the final draft.
- Sydney Wynne performed the experiments, approved the final draft.
- Elizabeth W. Alm conceived and designed the experiments, performed the experiments, analyzed the data, contributed reagents/materials/analysis tools, authored or reviewed drafts of the paper, approved the final draft.

- Peter S. Kourtev conceived and designed the experiments, performed the experiments, analyzed the data, contributed reagents/materials/analysis tools, authored or reviewed drafts of the paper, approved the final draft.

## DNA Deposition

The following information was supplied regarding the deposition of DNA sequences:

National Center for Biotechnology Information's (NCBI) Short Read Archives (SRA), accession number: SRP120551.

## Data Availability

The raw data are contained in the Methods section of this article and the Supplemental Materials.

## Supplemental Information

Supplemental information for this article can be found online at http://dx.doi.org/10.7717/peerj.6258#supplemental-information.

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
