# Peer review of "Comparative genomics of 16 Microbacterium spp. that tolerate multiple heavy metals and antibiotics"

_PeerJ, doi:10.7717/peerj.6258_

## Round 0.1 · original submission · Major Revisions

All reviewers and myself found your work quite interesting and found merit deserving publication. However there are several aspects a on it that could be improved substantially increasing the impact of the work and the message. I encourage you to follow all the comments and suggestions provided, specifically:

-Improve the message of the abstract and the overall results and conclusions achieved.

- Focus more in the introduction about your hypothesis and objectives and why you specifically studies this bacteria genera.

- Try to improve clarity of the information probvided in the figures and include some extra-supplementary ones as indicated

- I consider that your data can also be more fully explored adding additional analyses as reviewer 2 suggested. It is obvious that a phylogenetic analysis based only in 16S in not enough and since you have the full genomes much more can be done

·

Basic reporting

no comment

Experimental design

no comment

Validity of the findings

no comment

Additional comments

Due to their high genetic diversity and adaptation ability, bacteria play an vital role in bioremediation of heavy metal pollution. This manuscript reported the tolerance of 16 strains of Microbacterium spp. to heavy metal and antibiotics, and many genes related to Cr(VI) reduction, heavy metal tolerance and antibiotic resistance were identified. In general, this manuscript merits to be published in Peer J.
1. Line 26, Since "Sixteen different strains....." was mentioned at the beginning of Abstract, ANI among the sixteen isolates was suggested to provide.
2. Line 89, what was criterion to select the 16 isolates? Were they selected randomly?
3. Line 124-125, Since 16S rRNA gene is duplicate gene, was it identical in one strain? which one was selected to generate phylogenetic tree?
4. A phylogenetic tree on the basis of chrR gene is suggested to provided.
5. "sp." should not be italic in table1-table3.
6. It is interesting that two concentrations (0.5 mM and 1 mM) were adopted to the ability of Cr reduction. Why?

·

Basic reporting

The manuscript is well written, with few typos and unclear sentences (see general comments) and I was able to access all data as shared/reported.

The introduction does a great job outlining chromium tolerance, and the genes associated with these tolerances, but does not explain what comparative genomics approach adds to the knowledge regarding heavy metal resistance or Microbacterium diversity. Since comparative genomics is the primary method/objective for this study more of the introduction should define and detail what this will provide. Discussions of ecotypes (Hunt et al., 2008 Science; Denef et al., 2010 PNAS; Frangeul et al., 2008 BMC Genomics; Briand et al., 2009 ISME J) and known diversity within the Microbacterium would be a potential place to add needed context (Meyer et al., 2017 PLoS ONE). Additionally, a few sentences identifying the specific knowledge gap this study (or these isolates) fills would provide better context at this point in the manuscript. The nature (metals, sources) of contamination of the study site is not described, which would provided the context for such gaps in knowledge as: Is reduction a proxy for “cleaning up” this environment? Are heavy metals a continued issue, creating a level of steady-state heavy metal stress? Are Microbacterium found in similar contaminated environments in other areas so the knowledge here could inform elsewhere?

I really enjoy figures like Figure 1 because they contain so much information, however, not all readers will be as familiar with An’vio and the way data are presented. If you could expand on the description of what data are contained, and where they are displayed in the figure, more readers would be able to fully explore all the information presented in Figure 1. For instance, what do the dark vs. light portions of the genome plots mean? What does the outermost ring show? Could a scale/legend be added to the tree above the chromate reduction and antibiotic genes?

I was only able to open figure S1 using Adobe Photoshop. Some readers may not have the same access to such software, please consider changing the file format to a more universal type so that all readers can easily view this figure.

Would it be possible to add an additional figure to show the entire pathway, and genes for each step, for the functions discussed in this manuscript? For example in Flores and Herrero (2005 Biochem. Soc. Trans.) Figure 1 helped to show the variety of Nitrogen assimilation and reduction pathways in cyanobacteria. A similar figure in this manuscript could help to show how the chrA and chrR genes would function in relation to one another or independently.

Experimental design

The primary research objective was not entirely clear to me, given that Microbacterium resistant to chromium were previously isolated and studied in this environment (Henson et al., 2015). The inclusion of comparative genomics in the title combined with the focus on heavy metal and antibiotic resistances in the introduction suggest that more focus will be placed on shared (core) and differences (flexible) in genome content. Such differences can better inform population diversity and function and would be filling a knowledge gap. Additionally, Lines 212-214 suggests that “other Microbacterium spp.” were not included the comparative genomics analysis. Why not? I agree that the identification of chrR and yeiF homologues in these other isolates suggest high interspecies diversity, and that diversity would allow for better calculations of core and pan-genome sizes in addition to providing a more complete picture of metal and antibiotic resistances.

The CheckM results suggest that some isolates contain strain heterogeneity, which may be an artifact of mis-assembly of closely related organisms. Hug et al. (2015, Environmental Microbiology) determined that assembly was improved for more abundant organisms’ genomes when subsampled, but the level of subsampling depended on the sample. Testing multiple subsample depths (sequential assembly) could reduce strain heterogeneity in some of your assemblies and could identify multiple isolates from the same sample/culture.

It is not clear if the flexible genome was specifically searched for differences and resistance/tolerance genes (using homology, BLAST, etc.). Perhaps you could add a paragraph outlining the major COG categories of the flexible genome; most will be unknown/hypothetical, but those that have an assigned function could be very interesting as well.

Given that this was a whole-genome approach, I was surprised that the phylogenetic tree was only based on 16S rRNA sequences and did not include Microbacterium isolates mentioned in the text (Lines 183-189). If available these other isolates should be included in the tree as it may highlight subclades with shared function. The use of whole genome gene presences/absence should also be considered to make a more robust phylogenetic tree that takes into account difference in gene content that the authors have access to through their sequencing efforts (as observed in Humbert et al., 2013 PLoS ONE; Meyer et al., 2017 PLoS ONE).

While IMG is a great tool for exploring genomes, it was not clear to me exactly how searches were performed within IMG (Lines 128-135). If using a keyword search, some genes may have different annotations or missing keywords and therefore not show up in search results. Was the COG search a secondary search to verify that all target genes were found or a separate search entirely? If the COG search was separate, please consider double checking that all chromate resistance & reduction and heavy metal resistance genes were found through secondary searches that do not rely on keywords in the annotation.

Please include a description of how you calculated gene homology in your methods. Were the BLAST comparisons bi-directional? Table S3 suggest these were one-way blast comparisons. What informed your decision in choosing cutoff values for homology? See Albertsen et al., 2013 Nature Biotechnology for such a comparison. Did the Microbacterium chrR genes have a higher homology to one another than to the Pseudomonas reference? I would like to see the data on the interspecies genetic diversity of chromate reductases. Perhaps a table or figure depicting the homology of these specific genes.

Validity of the findings

Your experiments did a good job of testing for and identifying potential genes involved in metal resistance, and then verifying resistance at a range of exposure concentrations. Useful context would be to briefly discuss observed concentrations of the tested metals in natural and contaminated environments. Levels considered to be “contaminated” are not discussed but would be useful in interpreting the different strains resistance to the range of concentration tested. You do a great job provide this context for antibiotics in Lines 245-257.

I like how the introduction outlines the difference in function between Cr(VI) transport via ChrA and Cr(VI) reduction via chrR and yieF and would like the discussion to also touch on this difference since no chrA were identified in your isolates (“Implications to community functions” would be the ideal paragraph). Does this strategy (reduction only) have any broader implications for not only population differentiation, but community composition and function?

You did a great deal of work, sequencing and comparing 16 genomes, and I'd like to see more discussion on differences in gene presence/absence, or even differences in annotated categories of genes (COGs, KEGGs, etc.) between the core and flexible genome. These data will potentially allow for more conclusions on the variation and genetic diversity of the community. It would also lend more weight to the speculation between genomics and physiological diversity and differentiation within this environment.

Additional comments

Multiple In-text references missing in Reference list. For example, Brose and James, 2010; Meyer and Huber, 2014. Please check all references for inclusion and accuracy.

The use of italics for gene abbreviations (ex: chrA) is widely accepted, please double check that all gene abbreviations are italicized throughout the text.

Table 1: Footer ** says “accept for” should be changed to “except”

Lines 75-80: These statements are valid, but seem more suited for the abstract or discussion.

Line 111: Why was 70bp chosen as the new read length for secondary quality filtering?

Lines 122-123: How was Prodigal used to annotate the core genomes? Prodigal provides gene prediction and not annotations; this method description seems incomplete unless “annotation” simply means marking their location(s) across the isolates compared.

Lines 130-132: Was this a nucleotide or amino acid BLAST? Bi-directional BLAST or one-way? Were these BLASTs done within the IMG/MER interface or via command line?

Lines 132-134: At first read the phrase “…were also annotated using KEEG pathway KO identifiers” makes it seem that these genes were annotated by the authors. Was this meant to say that these genes were identified/searched for using KEGG (I believe KEEG is a typo) and KO identifiers? I think just replacing “annotated” with “identified” would be sufficient if that was the intent.

Lines 165-167: I agree that this is a very interesting finding and would like to see this expanded in the discussion. Are there other reports of similar observations?

Line 190: I read the sentence two possible ways: 1- the core genome has a large amount of variation, meaning that shared genes are significantly different among the 16 isolates or 2- the pangenomic analysis succeeded in identifying the core genome and separately found regions that were more variable (flexible genome). Would it be possible to clarify this either though 1- adding metrics for the variation in core genes (SNPs, phylogenetic trees using just the Core or non-core genes) or 2- Removing this sentence as the following line is a great summary of the pangenome initial results for core/pan genes.

Lines 192-193: While this reference is valid, it is also very general. The Bacillus examples presented are from a different phylum. Are there any estimates for core-genome size/percentages for Microbacterium or higher taxonomic levels (Actinobacteria) that could provide better context for whether the 26% calculated in this study is typical for related organisms? Alternately, what about the percent core genome for other organisms from similar environments? Is a larger flexible genome an adaptation for heavy metal exposure? I appreciate this line of thinking but wanted it to dive deeper.

Line 248-249: Awkward sentence. Errant comma after “study” and use of “found” twice seems repetitive. Suggest “A 2015 study found 26% of Microbacteriums isolates from ____ (environments) to be resistant to vancomycin.”

Lines 247-257: You list the three antibiotics in Line 247-248 and have them ordered in the same manner in Table 3 (amp, can, van). However, in Lines 248-254 they are listed in different orders each time. I suggest re-structuring these discussion points to keep that order and discuss one antibiotic at a time unless a relationship between environment/exposure and isolate is more pronounced.

Lines 252-254: This is a very interesting comparison, if only because the tolerances in fish mucus are so much higher. Why so high in fish? What sort of exposure concentrations are typical for fish mucus? Are these isolates included in your phylogenetic tree? How similar are the

Line 254-257: Specific details for the Microbacterium isolated in this study are provided for vancomycin and chloramphenicol, but not ampicillin (even though Isoalte PF3 had a relatively high resistance compared to other isolates). Please include all three results.

Reviewer 3 ·

Basic reporting

The language used is mostly unambiguous and professional. I have have commented for parts where it is ambiguous in general comments.
Literature references are sufficient but I have suggested some more.

Experimental design

Research question is not well defined. Objective of the study is not clear. It should be mentioned at least in the end of Introduction section.

Validity of the findings

Data is well analyzed and interpreted. There are many conclusions drawn in the Discussion section, However, authours need to conclude their abstract strongly as well.

Additional comments

The manuscript describes the isolation and genome analysis of 16 multi-metal and antibiotic resistant microbacterium species. The authours have determined the metal and antibiotic resistance of the isolates and have sequenced the genomes of the isolates using Illumina Hiseq sequencing. The authours have analysed these genomes looking for presence of possible metal and antibiotic resistance genes and mechanisms and have also done pangenomic analysis. The work described is satisfactory, however, there are certain things that could be make it more interesting and compelling.

General Comments:
Authours should provide a list of abbreviations used in the article.
Cr has been mentioned as Cr (VI) (containing space) at some places and as Cr(VI) elsewhere. Authours should use only one format.
μg/ml should be μg ml-1

Major comments:
Why only Microbacterium species were isolated and focused for this study? The authours should describe it and add related literature in the introduction as well.
Line 32-34: The authours have ended the abstract saying “…was likely required of this population…”. It is already a known fact that certain bacteria/ the bacteria that survive certain toxicants are able to do it due to the presence of such genes and mechanisms. I believe the authous should and can come up with a better conclusion to the abstract. The authours should mention what different they have found in their study and what they can conclude out of it.
Line 74-80: Here instead of summarizing the study, the authours should mention objective of the study.

Minor comments:
Abstract:
Line 25: (Cr[VI]) should be [Cr(VI)]
Line 26: “…were able to tolerate high concentrations (0.1 mM) of cobalt, cadmium, and nickel…”. These are not high concentrations as much higher concentrations have been reported. Moreover, authours should mention chromate tolerance level of the isolates here in the abstract as well here.
Line 28: Authours should mention the concentration of the antibiotics resisted.
Introduction:
Line 46: “Bacteria are able to tolerate Cr(VI)…” should be “Many bacteria are able to tolerate Cr(VI)…” as not all bacteria can tolerate Cr(VI).
Line 58: “…vital role in how bacterial thrive…” should be “…vital role in how bacteria thrive…”
Line 79: “…within microbial populations may help…” should be “…within microbial populations that may help…”
Methods:
Line 87: Was the isolation site already known to have high chromium and other metals concentration?
Line 94: “…with 0.5 mM or 1 mM Cr (VI)…”. Were two concentrations used? Kindly explain.
Line 94 and Line 99: The authours here have said that the isolates were grown with 0.5 mM or 1 mM Cr (VI), and with 0.5 mM or 2 mM, respectively. However, in Line 90 the authours stated “Sixteen isolates were further selected based on their ability to grow on 0.5 mM Cr(VI).” Kindly clarify why they used 1.0 and 2.0 mM concentrations when the maximum tolerance level of chromium checked was 0.5 mM.
Line 104: The auhours should mention how may Hiseq flowcells they used for sequencing. Moreover, in the relevant results they should also mention the coverage they obtained.
Line 128-130: Not clear. Kindly modify.
Line 132: Why the authours decided to compare chrR with that of Pseudomonas putida KT2440?
Line 139-140: Metal tolerance is best determined using broth cultures. Moreover, 10% means 10% nutrients or agar, or both?
Results and Discussion:
Line 160-163. The authours stat that they had 16 isolates showing varied Cr resistance. But here it is confusing as authours are saying that 13 isolates were able to tolerate 2mM Cr(VI), 3 were able to tolerate only 0.5mM Cr(VI), and 5 isolates were able to tolerate 20mM Cr(VI). Kindly clarify here.
Line 164: I think its better to say “…over a 72-hour incubation” rather “…over a 72-hour experiment”.
Lin 175-176: “…all had GC content over 68%”. GC-content is a major character for species identification and classification. Authours should comment on the variability of the GC-content they found in the genomes of their isolates.
Line 200-202: The authours have confirmed the resistance of the isolates against metals and antibiotics through lab experiments. Therefore, the use of the word “may” here is confusing.
Line 207-208: Repetition of chrR ?
Line 243: “Thus, is it possible the presence of these genes…” should be “Thus, it is possible that the presence of these genes…”
Line 261-67: The authours should also cite here doi:10.1080/01490451.2016.1240265 and http://dx.doi.org/10.1016/j.crvi.2017.05.002.
Line 264: “…for metals and antibiotics was found Staphylococcus aureus...” should be “…for metals and antibiotics was found in Staphylococcus aureus...”
Line 280: “All Microbacterium spp. isolates were isolated from…” should be “All Microbacterium spp. were isolated from…”
Line 284-87: the authours should also cite doi:10.1111/jam.13535 for genotypes of Klebsiella pneumoniae, Bacillus subtilis, and Citrobacter freundii.
Line 293: I think “…stressors (e.g. heavy metals) promoted genomic diversity…” should be “…stressors (e.g. heavy metals) can promote genomic diversity…”
Table 1: Instead of giving * the authours can mention mM alongside Metal tolerance (mM).
Table 3: * should go with Cam and Van as well.

---

## Round 0.2 · Minor Revisions

There are a few small issues mainly of misspelling or clarification that need to be addressed before final acceptance. Thanks

·

Basic reporting

Edits to the introduction have provided context and objectives previously lacking.


Paragraph: "Bacteria are also know to catalyze the reduction..." ~Line 65
Anaerobic vs. anaerobic reduction of Cr(VI) is not mentioned again in the manuscript. Given the new context, it is not clear to me that discussion of anaerobic reduction of Cr(VI) is necessary. Was it hypothesized that the Microbacterium were using anaerobic pathways? Was the study site hypoxic/anoxic? I can see how demonstrating a diversity in known Cr(VI) reduction pathways is important, but would like to see the aerobic/anaerobic topic to be re-visited in the discussion, if only briefly.

I appreciate the additional text to describe the figures.

Experimental design

Response to specific comments:
Review: The CheckM results suggest that some isolates contain strain heterogeneity, which may be an artifact of mis-assembly of closely related organisms. Hug et al. (2015, Environmental Microbiology) determined that assembly was improved for more abundant organisms’ genomes when subsampled, but the level of subsampling depended on the sample. Testing multiple subsample depths (sequential assembly) could reduce strain heterogeneity in some of your assemblies and could identify multiple isolates from the same sample/culture.

Response: During our assemblies, we tested different subsamples of our reads. In the end, our assemblies with 4 million reads provided the highest quality assemblies. Text has been added to mention that multiple depths of subsampling were tested.

Re-review: Additional text addresses my initial comment, thank you.
* * *
Reviewer: It is not clear if the flexible genome was specifically searched for differences and resistance/tolerance genes (using homology, BLAST, etc.). Perhaps you could add a paragraph outlining the major COG categories of the flexible genome; most will be unknown/hypothetical, but those that have an assigned function could be very interesting as well.

Response: We have added data to show the flexible genome, as it was not previously assessed. We have added additional text about the variability but we kept it focused on metals and antibiotic tolerance as we feel a strength of our work is that it couples both genomics and physiological data.

Re-review: I appreciate the additions and agree the authors place a strong focus on coupled genomics and physiological data.
* * *
Reviewer: The use of italics for gene abbreviations (ex: chrA) is widely accepted, please double check that all gene abbreviations are italicized throughout the text.

Response: We initially tried to use “chrA” for text relating to genes and then “ChrA” when referring to the function, however, for consistency we have changed all to “chrA”.

Re-review: I have no issue with the different formatting being used to differentiate genes vs. functions, unfortunately this was not initially clear to me. I appreciate the authors’ willingness to change all to “chrA” for consistency but did notice many instances of non-italicized gene abbreviations in the Introduction (Ex: lines 68-69).
* * *
Other comments have been addressed and been made more clear in the text.

Validity of the findings

Additional commends regarding core and flexible gene content are very informative and I think add to the strength of your paper and highlight the amount of analysis done to produce this data set.

Additional comments

Line 41: “…anthropogenic activities, such as industrial and mining industries” is awkward. Is mining industries an activity? Perhaps a statement “such as industrial manufacturing (items that need heavy metals) and mining (mine types).”

Line 42: Missing period after parentheses. Change “exists” to “exist.”

Line 56: “stressors” not “strossors”

Line 134: Illumina misspelled as “Illumnia.”

Lines 149-153: “A pangenomic analysis…” After edits the sentence reads awkwardly. I understand the intent and context of the edits but recommend re-writing this sentence to improve clarity. Perhaps: “A pangenomic analysis of 20 genomes (from this study and Henson et al., 2015) was conducted using…” or similar.

Line 159: “The IMG systems …of the isolates.” This sentence seems out of place since it is followed by outlining how sequences were downloaded and run through MEGA7. Recommend this sentence is just removed given that the IMG searches are well outlined later in the paragraph.

Line 280: "All did have an annotated..." Suggest, "All possess"

Reviewer 3 ·

Basic reporting

The authors have made sufficient and necessary modifications to the manuscript and is in acceptable form.

Experimental design

The authors have made sufficient and necessary modifications to the manuscript and is in acceptable form.

Validity of the findings

The authors have made sufficient and necessary modifications to the manuscript and is in acceptable form.

Additional comments

The authors have made sufficient and necessary modifications to the manuscript. I recommend the manuscript for publication. It has been an honour to review it.

---

## Round 0.3 · accepted · Accept

Thanks for final modifications of the manuscript. Your manuscript is ready for publication now.

#